# Two-Month Consumption of Orange Juice Enriched with Vitamin D3 and Probiotics Decreases Body Weight, Insulin Resistance, Blood Lipids, and Arterial Blood Pressure in High-Cardiometabolic-Risk Patients on a Westernized Type Diet: Results from a Randomized Clinical Trial

**DOI:** 10.3390/nu16091331

**Published:** 2024-04-28

**Authors:** Emilia Papakonstantinou, Nikolaos Zacharodimos, Georgios Georgiopoulos, Christina Athanasaki, Dionysia-Lydia Bothou, Sofia Tsitsou, Foteini Lympaki, Stamatia Vitsou-Anastasiou, Olga S. Papadopoulou, Dimitrios Delialis, Evangelos C. Alexopoulos, Eleni Petsiou, Kalliopi Keramida, Agapi I. Doulgeraki, Ismini-Maria Patsopoulou, George-John E. Nychas, Chrysoula C. Tassou

**Affiliations:** 1Laboratory of Dietetics and Quality of Life, Department of Food Science and Human Nutrition, School of Food and Nutritional Sciences, Agricultural University of Athens, 11855 Athens, Greece; nikolaoszacharodimos@gmail.com (N.Z.); chrathanasaki@gmail.com (C.A.); lydiampothou@gmail.com (D.-L.B.); stsitsou@aua.gr (S.T.); fotinilyb28@hotmail.com (F.L.); 2Department of Clinical Therapeutics, Alexandra Hospital, National and Kapodistrian University of Athens, 80 Vasilissis Sophias Ave, 11528 Athens, Greece; georgiopoulosgeorgios@gmail.com (G.G.);; 3Institute of Technology of Agricultural Products, Hellenic Agricultural Organization “DIMITRA”, 1, S. Venizelou, Lykovrissi, 14123 Attiki, Greece; matinavitsou@outlook.com.gr (S.V.-A.); olgapapadopoulou@elgo.gr (O.S.P.); ctassou@elgo.gr (C.C.T.); 4Laboratory of Microbiology and Biotechnology of Foods, Department of Food Science and Human Nutrition, School of Food and Nutritional Sciences, Agricultural University of Athens, 75 Iera Odos, 11855 Athens, Greece; gjn@aua.gr; 5Ergomneia Medical PCC, Ellispontou 11, 15669 Papagos, Greece; ecalexopoulos@hotmail.com; 6Henry Dunant Hospital, 107 Mesogeion Ave, 11526 Athens, Greece; dr.petsiou@yahoo.gr; 7Cardiology Department, General Anticancer Oncological Hospital Agios Savvas, 171 Alexandras Ave, 11522 Athens, Greece; keramidakalliopi@hotmail.com; 8Laboratory of Food Microbiology and Hygiene, Department of Food Science and Technology, School of Agriculture, Aristotle University of Thessaloniki, 54124 Thessaloniki, Greece; adoulgeraki@agro.auth.gr; 9Shandong Agricultural University, Tai’an 271002, China

**Keywords:** fruit juice, vitamin D, probiotics, blood glucose, insulin resistance, blood lipids, blood pressure, body weight, adults

## Abstract

This study examined the effects of orange juice (OJ) supplemented with vitamin D3 (2000 IU) and probiotics (*Lacticaseibacillus casei* Shirota and *Lacticaseibacillus rhamnosus* GG, 10^8^ cfu/mL) on cardiometabolic risk factors in overweight and obese adults following a Westernized-type diet. Fifty-three high-risk individuals were randomly assigned to one of two groups. Over 8 weeks, one group consumed a vitamin D3 and probiotic-enriched OJ and the other regular OJ (control). Diets remained unchanged and were documented through food diaries. Measures of metabolic and inflammatory markers and blood pressure were measured at the start and end of the study. Post-intervention, the enriched OJ group showed the following significant metabolic improvements (without changes in triglycerides, inflammation, or central blood pressure): reduced fasting insulin, peripheral blood pressure, body weight (−1.4 kg 95% CI: −2.4, −0.4), energy (−270 kcal 95% CI: −553.2, −13.7), macronutrient (dietary fat −238 kcal 95% CI: −11.9, −1.0; carbohydrates −155 kcal 95% CI: −282.4, −27.3; sugars −16.1 g 95% CI: −11.9, −1.0) intake, and better lipid profiles (total cholesterol −10.3 mg/dL 95% CI: −21.4, 0.9; LDL-C −7 mg/dL 95% CI: −13.5, −0.5). The enriched OJ led to weight loss, less energy/macronutrient consumption, improved lipid profiles, and increased insulin sensitivity after 8 weeks in those following a Westernized diet, thus indicating potential benefits for cardiometabolic risk. This study was a part of FunJuice-T2EDK-01922, which was funded by the EU Regional Development Fund and Greek National Resources.

## 1. Introduction

The exponential prevalence of overweight and obesity worldwide necessitates the development of effective strategies to address this escalating public health challenge. Among various interventions, dietary modification remains a cornerstone for managing the metabolic sequelae of increased adiposity. Obesity is a complex condition involving glucose dysregulation, lipid abnormalities, and systemic inflammation, which increases the risk of metabolic syndrome, insulin resistance, type 2 diabetes, and cardiovascular disease. To address such multifaceted physiological disruptions, researchers have turned their attention to innovative dietary interventions that leverage the combined effects of bioactive substances, including vitamin D and probiotics. Vitamin D3 has emerged as a focal point in enhancing insulin sensitivity [1], modulating lipid profiles [2], and exerting anti-inflammatory effects [3], while probiotics have been recognized for their ability to modulate the gut microbiota [4], systemic inflammation [5], and lipid metabolism [6]. A recent systematic review of seven clinical randomized controlled trials (RCTs) underscored the synergistic benefits of vitamin D and probiotic co-supplementation (lactic acid bacteria) in ameliorating chronic disease markers and improving metabolic health more effectively than either component alone [7]. Despite these advances, the collective impact of these bioactive compounds, especially when vitamin D3 is combined with the probiotic strains *Lacticaseibacillus casei* Shirota and *Lacticaseibacillus rhamnosus* GG in functional foods such as orange juice (OJ), is yet to be comprehensively investigated.

OJ remains the leading choice for 100% fruit juice worldwide because of its sensory and nutritional appeal. Although the health benefits of fresh fruits are well documented, the role of fruit juices in health outcomes remains controversial. Fruit juice can elevate postprandial glucose levels, prompting recommendations to limit its consumption. Nonetheless, evidence indicates that moderate daily intake (75–224 mL or greater, up to 500 mL) does not exacerbate chronic disease risks and may offer vascular, anti-inflammatory, and antioxidant benefits [8].

Our previous work explored the acute metabolic effects of biofunctional components (vitamin D3 or *n*-3 polyunsaturated fatty acids), probiotics (*Lacticaseibacillus casei* Shirota and *Lacticaseibacillus rhamnosus* GG), and their combination (vitamin D3-*n*-3-probiotics) in mixed fruit juices, and it also revealed that these ingredients might modulate glycemic and insulin responses, as well as influence appetite regulation differently across juice types in healthy individuals [9]. We also observed that preloading vitamin D3, *n*-3 fatty acids, and probiotic-enriched OJ could enhance satiety and reduce subsequent energy intake, particularly in overweight/obese individuals [10].

This study aimed to extend this research by evaluating the short-term effects of OJ enriched with vitamin D3 and probiotics on cardiometabolic risk factors in adults adhering to a Westernized diet. A comprehensive examination of anthropometric data, biochemical markers, and blood pressure (BP) pre- and post-intervention is proposed to elucidate the metabolic impacts of this functional food intervention.

## 2. Materials and Methods

### 2.1. Participants

A total of 53 adults with high cardiometabolic risk were enrolled in this randomized, double-blind, and parallel clinical trial. Recruitment was conducted through various channels, including referrals from medical practitioners, online platforms, and announcements on university premises. Eligible participants, based on the specified biomarker thresholds justified by recent guidelines, had a body mass index (BMI) > 25 kg/m^2^, were aged 18–70, and presented with hyperglycemia (fasting blood glucose > 100 mg/dL) and/or hyperlipidemia (total cholesterol > 200 mg/dL, LDL-C > 100 mg/dL, HDL-C < 40 mg/dL for men and <50 mg/dL for women, triglycerides > 150 mg/dL) and/or hypertension (systolic blood pressure > 130 mmHg or diastolic blood pressure > 80 mmHg). The exclusion criteria included medication known to affect glycemia (metformin, glucocorticoids, glucagon-like peptide-1 agonists, and thiazide diuretics); use of probiotics or antibiotics for at least two months before the study; consumption of >40 g alcohol per day; participation in weight reduction programs or other dietary interventions during the previous six months before study participation; diabetes mellitus; cardiovascular or coronary heart conditions; liver disease; kidney disease; gastrointestinal disorders; severe depression; and any chronic disease that might influence inflammatory markers. The female participants were not pregnant or lactating. Before study participation, all volunteers underwent an initial screening that included medical examination; medical history; sociodemographics; diet history; detailed biochemical examination (within six months before study participation); anthropometry (body weight, height, and waist and hip circumferences); body composition analysis using the bioimpedance method (InBody 230, Biospace, Cerritos, CA, USA); fasting blood glucose via finger prick (calibrated MediSmart^®^ Ruby glucose meter with lancing device, Lilly Pharmaserv, SA, Athens, Greece); and blood pressure using an upper blood pressure monitor (Omron, Intellisense, HEM-907, Omron Hellas, Athens, Greece). Participants receiving antihypertensive and hypolipidemic treatment were included in the study. Antihypertensive treatment included irbesartan, manidipine, amlodipine, ramipril, nebivolol, and candesartan. Antihyperlipidemic treatments included rosuvastatin and pitavastatin. Participants were asked to maintain a constant dose and treatment type throughout the study.

This study was conducted at the Laboratory of Dietetics and Quality of Life, Agricultural University of Athens, Greece. All participants provided informed consent for inclusion in the study. This study was conducted in accordance with the guidelines detailed in the Declaration of Helsinki, and the protocol was approved by the Bioethics Committee of the Agricultural University of Athens (EIDE Reference Number 75 4 October 2022). This trial was also registered at Clinicaltrials.gov (NCT06114576).

### 2.2. Study Design

The participants were stratified by age, sex, and antihypertensive/hypolipidemic treatment before randomization to ensure group comparability. Random allocation to the intervention or control group was conducted using secure online computer software (Social Psychology Network, Middletown, CT, USA) (http://www.randomizer.org (accessed on 15 January 2022) [11] by an independent researcher who was not involved in the collection and analysis of the study’s data. Double-blind conditions were achieved, in which both the researchers and participants were unaware of the group assignments. The final number of participants was 26 in the intervention group (no withdrawals) and 24 in the control group (after accounting for withdrawals). The study design (Figure 1) details the flow of the participants through the trial.

Each participant was instructed to consume 250 mL of their assigned OJ (Aspis SA Hellenic Juice Industry, Argos-Korinthos, Greece) daily and 30 min before their lunchtime meal. The OJ consumed in the intervention group was fortified with 2000 IU vitamin D3 and 10^8^ cfu/mL probiotics. The control group received unfortified OJ. All participants were required to maintain their habitual diet throughout the 2-month OJ consumption period.

Weekly visits including anthropometric measurements and dietary assessments were conducted. Participants visited the Laboratory of Dietetics and Quality of Life weekly to receive a packet (cool box) of 7 OJ portions (enriched or plain) in quantities of 250 mL each to consume daily. At the first appointment, with a dietitian of the team, participants were informed about the study protocol and were asked to keep a 3-day food diary. Participants were asked to record the juice serving and foods and beverages they consumed, including the time and location of consumption and any additional notes that might be relevant, such as feelings of satiety or reasons for missing a serving or meal. During the 8-week intervention period, each group of participants followed their habitual Westernized-type diet without any further consultation. The intervention group (10 men and 16 women) received 250 mL of OJ enriched with vitamin D3 and encapsulated probiotics daily, which provided 2000 IU of vitamin D3 and 10^8^ cfu/mL of probiotics per day. The control group (7 men and 17 women) received 250 mL of the same OJ as the intervention group daily, albeit without vitamin D3 and probiotic enrichment (control), along with their habitual diet. Participants in both groups were instructed to consume 250 mL of OJ (enriched or plain) before their lunch meal along with their habitual diet. Compliance with the daily drinking of the allocated juice and maintaining usual dietary and exercise habits was assessed using weekly food records and a physical activity questionnaire, as well as by scheduled telephone contact by the dietitians with the participants in the middle of the week. During these check-ins, food diaries were reviewed, and any discrepancies or omissions were discussed. Participants were given seven portions of OJ weekly and asked to return any unopened portions at the end of each week. No unopened OJ was returned at any time point.

Body weight, waist and hip circumferences, and body composition were measured for all participants at the beginning of the study and weekly throughout the study. Basal metabolic rate (BMR; Ultima CPX, Medical Graphics UK Limited, Gloucestershire, UK) was measured at baseline, as well as at 4 and 8 weeks. Both groups were closely supervised by the dietitians and nutritionists of our research team, who made weekly phone calls and scheduled appointments with the participants. During these sessions, arthrometric measurements were performed for all participants, who also handed out their 3-day food diary.

### 2.3. Encapsulation of the Microbial Cultures and Orange Juice Inoculation

The probiotic microorganisms *Lacticaseibacillus casei* Shirota (ACA-DC 6002) and *Lacticaseibacillus rhamnosus* GG (ATCC 53103) were used in this trial, as was previously described in [9]. Briefly, the strains were revived from a stock culture at −80 °C and cultured twice in a fresh-liquid MRS growth medium (MRS broth, 4017292, Biolife, Milano, Italy) for 24 h at 30 °C. The monocultures were then centrifuged (5000× *g*, 10 min, 4 °C), the supernatant was discarded, and the final pellet of the two strains was resuspended in a ¼ strength Ringer’s solution (Ringer solution Tablets, 96724-100TAB, Merck, Darmstadt, Germany). This was then mixed at a 1:1 ratio to a final concentration of 10^10^ cfu/mL.

The two strains were encapsulated as previously described by Bosnea et al. (2017) [12] with minor modifications. Aqueous solutions of powdered whey protein isolate (3% *w*/*w*) (WPI BiproTM, 92.08% *w*/*w* protein, 1.08% *w*/*w* fat, 4.08% *w*/*w* ash, 1.08% *w*/*w* lactose, Davisco Foods International Inc., Le Sueur, MN, USA) and gum arabic (3% *w*/*w*) (GA, Sigma Chemicals, Gillingham, UK) were prepared using ultrapure water (HPLC grade), and the solutions were gently stirred (for 3 h at 210 rpm) (Orbital and Linear Digital shaker, RS Lab/RSLAB-7, RSLab, Heraklion, Crete, Greece) at room temperature. The solutions were stored at cold temperature (4 °C) for 24 h for complete solubilization and hydration. To prepare the WPI:GA coacervate, aqueous solutions of whey protein isolate and gum arabic (WPI:GA) were mixed at a 2:1 ratio. This final solution was inoculated with the probiotic strains under continuous agitation, and food-grade citric acid (10%) was added to lower the pH to 4. Then, 10 mL of this solution was added to sterilized breast milk bags (volume of 250 mL), which were used as containers for the OJ. Finally, the mixture (coacervate) was left at room temperature for phase separation. After 1 h, the supernatant was discarded, and the resulting precipitate constituted the WPI:GA microencapsulation system, which contained the two probiotic microorganisms. To produce the juice samples, OJ (250 mL) and vitamin D (2000 IU) (dry vitamin D3 100 GFP SD, BASF, Ludwigshafen, Germany) were added to breast milk bags containing the encapsulated probiotics at a final inoculum concentration of 10^8^ cfu/mL (enriched OJ). The selected population (10^8^ cfu/mL of each strain) fulfilled the criteria for probiotics to provide health effects, where the health effects of probiotics were dependent on dose, and a consumption of 10^9^ cfu per day was the minimum recommended dose [13,14]. 

The addition of vitamin D and encapsulated probiotics did not alter the organoleptic characteristics of the product. Vitamin D is colorless and odorless, and its addition to the OJ cannot be detected. In addition, several sensory assessments (by a trained panel of the Institute of Technology of Agricultural Products, Lykovryssi, Greece) during cold storage of the OJs (control and FJ) were performed to ensure the palatability of the product and to investigate whether there was any impact on the taste or consistency. The encapsulated probiotics and vitamin D were not visible to the volunteers, and no deviation in the taste or palatability of the juices was observed. 

The OJ samples (enriched OJ and control) were prepared under aseptic conditions (inside a laminar flow cabinet) using sterile bags (250 mL sterile bags for breast milk) as juice containers. All reagents used were sterilized before use, and the personnel wore appropriate protective equipment to ensure aseptic processing. To measure the bioavailability of the probiotics in the final products, the encapsulated cells were released from the microcapsules by adding an appropriate amount of 5 N NaOH to increase the pH to 7.4, at which the biopolymers in the coacervate structures no longer interact electrostatically since they both carry negatively charged groups. Then, the probiotic population was determined by counting the viable cells present in the juice samples after sampling (10-fold dilutions in ¼ strength Ringer’s solution), and in the juices and pour plating on MRS agar (incubation at 37 °C for 48–72 h) (detailed information can be found in Bosnea et al. (2017) [12]. This procedure was performed immediately after juice preparation and during the 7-day cold-storage period. All samples were stored at 4 °C and distributed to volunteers on the same day of production under cold conditions.

### 2.4. Dietary Assessment, Physical Activity Assessment, and Subjective Appetite

Participants were asked to record the type and amount of all foods and beverages consumed 3 days each week (2 weekdays and 1 weekend day) before entering the study and throughout the intervention. These records were reviewed weekly by dietitians (via phone calls or live meetings). For each 8-week intervention, a 3-day food diary was used to assess compliance with the dietary intervention. Detailed instructions were provided on how to record the quantity of food consumed using food scales or with standard household weights and measures. The dietitians also checked the food diaries for any misreporting, and, when necessary, used food models and photographs to clarify discrepancies in portion sizes. Food diaries were analyzed using Nutritionist Pro^TM^ diet analysis software (version 7.9, Axxya Systems LLC, Redmond, WA, USA), with extensive modifications to the database to include new foods and recipes.

Weekly subjective appetite assessments were conducted using a 100 mm line visual analog scale (VAS). VAS scores ranged from not at all (0 mm) to extremely (100 mm), with, for example, neither hungry (0 mm), full (100 mm), or having a desire for food in the middle (50 mm) over the previous two weeks. The VASs were given in the form of a booklet, and with one scale per page [15]. Physical activity was assessed with the Greek version of the International Physical Activity Questionnaire (IPAQ) [16]. The IPAQ collected data on the self-reported physical activity in the previous week and analyzed the amount of time spent performing light, moderate, and high-intensity activities, as well as sleeping. The mean daily energy expenditure and physical activity levels (PALs) were estimated based on the metabolic equivalents (METs) for each type of activity.

Vitamin D intake was estimated from the 3-day food records at baseline, at 4 weeks, and at 8 weeks. Sun exposure was assessed using the Greek version of a validated questionnaire with the aim of evaluating the amount of sun exposure and parameters that could influence vitamin D skin synthesis [17,18] at baseline, at 4 weeks, and at 8 weeks. It included questions related to sun exposure for each of the four seasons (weekends and weekdays), exposure for the last 30 days, sunscreen use, and skin color. All respondents were Caucasians and reported pallidity and the ability to tan, with skin color being categorized according to skin pallor and ease of tanning as very light–fair skin; light; medium light; fairly dark; dark and tans easily; and very dark and tans very easily. The vitamin D cut-off chosen was 20 ng/mL, as values <20 mg/mL reflect inadequate (12 to <20 ng/mL) or deficient (<12 ng/mL) levels for bone and overall health in healthy individuals [17,19]. 

### 2.5. Blood Sampling and Laboratory Methods

Participants underwent blood sampling at the Laboratory of Dietetics and Quality of Life between 08:00–09:00 at the beginning and end of the 8-week study after a 10–12 h overnight fast and an avoidance of alcohol and vigorous exercise. All venous blood sample analyses were performed at the same laboratory located in central Athens: Bioiatriki (Bioiatriki Healthcare Group and Diagnostic Centers, Attiki, Greece). Plasma glucose, plasma insulin, urea, serum total cholesterol, LDL-C, HDL-C, triglycerides, high-sensitivity *C*-reactive protein (CRP), interleukin-6 (IL-6), and serum vitamin D levels were measured using a fully automated analyzer (Cobas Integra 800, Roche, Basel, Switzerland). Insulin resistance (Homeostasis Model Assessment, HOMA-IR) was calculated using the following formula: fasting insulin in μU/mL multiplied by fasting glucose in mg/dL and divided by 405 [20]. An HOMA-IR greater than 1.9 indicates early insulin resistance, and a value greater than 2.9 indicates significant insulin resistance [21]. Insulin sensitivity was estimated using the fasting glucose-to-insulin ratio (FGI). FGI values lower than 4.5 indicate insulin resistance [22]. The quantitative insulin sensitivity check index (QUICKI), which is a quantitative insulin sensitivity check index, was estimated using the following formula: 1/(log(fasting insulin μU/mL) + log(fasting glucose mg/dL)). A QUICKI of ≤0.35 indicates insulin resistance [23].

### 2.6. Blood Pressure (BP)

Peripheral systolic and diastolic BP were measured at the screening stage, and at 2, 4, 6, and 8 weeks of the study using an upper arm digital BP monitor in a quiet warm setting. The participants rested for 5 min in the supine position with their arms supported at the heart level, after which three BP measurements were taken by an already introduced member of the trained team to avoid the “white coat effect”, at 1 min intervals, with the three readings then being averaged. 

### 2.7. Pulse Wave Velocity (PWV) and Central Blood Pressure

PWV was measured at the beginning and at 4 and 8 weeks using a noninvasive device. After at least 10 min of supine rest in a quiet room with the temperature controlled at 20 °C to 25 °C, measurements were performed by trained research team members (NZ, CA, ST, and DLB). The carotid-femoral PWV, an established index of aortic stiffness, is usually calculated from measurements of pulse transit time and the distance traveled between two recording sites [24,25]. The equation used to calculate PWV was PWV = distance (m)/transit time (s). All measurements were performed using a validated noninvasive device (Complior Analyze, Alam Medical, France) that allowed online pulse-wave recording and the automatic calculation of PWV. At the end of the test, two different pulse waves were simultaneously obtained at two sites (at the base of the neck for the common carotid artery and over the right femoral artery) with two transducers. Distance was defined as (distance from the suprasternal nerve to the femoral artery)—(distance from the carotid artery to the suprasternal notch; coefficient of variation, 2.4% for two repeated measurements) [26].

Radial artery tonometry, utilizing the SphygmoCor System by Atcor Medical, was employed for acquisition and analysis of the aortic pulse waveform. Central blood pressure measurements, which provide prognostic insights beyond peripheral readings, serve as crucial endpoints in assessing interventions aimed at cardiovascular disease. Peripheral pressure waveforms were recorded at the radial artery using a handheld, high-fidelity tonometer from Millar Instruments, which was calibrated against arterial pressures measured at the brachial artery. Subsequently, aortic pressure waveforms were derived using the established generalized transfer functions. Analysis of the resulting aortic waveform enables the calculation of indices that primarily reflect arterial and aortic stiffness, as well as the intensity of the reflected waves. The parameters measured from the central aortic waveform included the following: (1) an augmentation index adjusted for a heart rate of 75 bpm, representing the percentage difference between the second and first peaks of the central aortic waveform in relation to the aortic pulse pressure, and (2) the central systolic and diastolic pressures [26,27].

### 2.8. Statistical Analysis

All continuous variables were tested for normal distribution using the one-sample Kolmogorov—Smirnov test and normal probability (Q—Q) plots. Continuous variables are presented as the mean ± standard deviation of the mean (SD), unless stated otherwise and skewed as the median (1st, 3rd quartiles), and the categorical variables are presented as absolute numbers and frequencies. Differences in the measured variables at baseline and at 8 weeks were evaluated by grouped comparisons using independent and paired sample Student’s *t*-tests (a Bonferroni-corrected *p* value < 0.05 was considered to indicate statistical significance) for normally distributed variables and the Mann–Whitney U test for skewed data. For categorical variables, Pearson’s chi-square tests were performed to determine the differences between groups. Next, we implemented a linear mixed model analysis to assess the differential changes in the variables of interest between the two groups across the study duration. Random intercepts were included in the mixed models to allow for random variability at the baseline, while an unstructured variance–covariance matrix was used. Linear mixed models were additionally used to control for age and sex (fixed effects). Sensitivity analyses were performed, controlling for additional potential confounders (i.e., BMI, medication use, fat mass, HOMA-IR, IPAQ, QUICKY, and IL-6). The multivariable model building was based on biological plausibility. Using data from a study reporting a 2-week mean ± SD change in an LDL-C of 8.5 ± 11% among participants who consumed a controlled, therapeutic lifestyle change diet [28], a sample of 25 participants in each group was expected to have 99% power to detect a 10% group difference in LDL-C change (i.e., an additional 10% reduction in the intervention group when assuming a common SD value of an LDL-C change of 11%). Statistical analyses were performed using SPSS software (version 20.0; SPSS Inc., Chicago, IL, USA). We deemed a value of 0.05 to indicate statistical significance. The interaction terms were considered significant when the observed significance was <0.1. All tests were two-tailed.

## 3. Results

The demographic and clinical characteristics were well balanced between the groups at baseline (Table 1), thus confirming successful randomization. Normally distributed variables were expressed as the means ± SEMs, and the categorical variables were expressed as absolute values (frequencies, %). P-values were obtained using the Pearson chi-square test for categorical variables, and an independent *t*-test was used for normally distributed variables.

### 3.1. Macro- and Selected Micronutrient Intakes at Baseline and at the End of 8 Weeks 

The energy, macro-, and micronutrient intake assessed from weekly 3-day food diaries did not differ between the two groups at baseline (Table 2). Energy, protein, carbohydrate, total fat, trans-fat, monounsaturated and polyunsaturated fat intake, as well as salt and sodium intake were significantly lower only in the vitamin D3 and probiotic-enriched OJ group (*p* < 0.05 for all) (Table 2, Figure 2A–C) at 8 weeks than at baseline. Sugar, vitamin C, and potassium intake were significantly higher only in the control OJ group (*p* < 0.05 for all) (Table 2) at 8 weeks than at baseline. Dietary fiber, saturated fat, and dietary cholesterol did not differ between the two groups at the end of 8 weeks compared to the baseline (Table 2). A significant interaction effect between intervention and time was found for changes in energy, dietary fat, carbohydrate, sugar, and monosaturated fat intake in favor of the intervention group, as well as in the potassium levels in the control group (*p* < 0.1 for all, Table 2, Figure 2A,B). These interactions did not change substantially after further adjustment for BMI, fat mass, medication use, HOMA-IR, IPAQ, QUICKY, and IL-6 levels (*p* < 0.1 for all).

No differences were found in the vitamin D intake from the 3-day diaries, self-reported sun exposure, or parameters influencing vitamin D skin synthesis between the groups at any time point (*p* > 0.05). Vitamin D status did not differ between the groups (adequate vitamin D levels: control group: n = 13; intervention group: n = 13; inadequate levels: control group: n = 8; intervention group: n = 8; deficient levels: control group: n = 3; intervention group: n = 5). Vitamin D status was positively correlated with BMI (rho = 0.030, *p* = 0.02).

### 3.2. Anthropometric Characteristics of the Participants at Baseline and at the End of the 8-Week Period

Body weight, BMI, and hip circumference were significantly lower at 8 weeks than at baseline in the vitamin D3- and probiotic-enriched OJ group (*p* < 0.05 for all, Table 3, Figure 2D). Body fat mass was significantly lower at 8 weeks than at baseline in both groups. Water mass, muscle mass, lean body mass, waist circumference, and waist-to-hip ratio did not differ between the two groups at the end of the 8-week period (Table 3). Compared with that at baseline, the BMR at the end of 8 weeks increased significantly only in the control group compared to the intervention group (Table 3). A significant interaction effect between intervention and time was found for changes in body weight, BMI, and hip circumference in favor of the intervention group after adjusting for age and sex (*p* < 0.1 for all; Table 2, Figure 2C). After adjusting for HOMA-IR, IPAQ, QUICKY, and IL-6 levels, these interactions did not markedly change (*p* < 0.1), whereas, after further adjustment for BMI and fat mass, these interactions were not significant (*p* > 0.1). Physical activity levels and subjective appetite scores did not change significantly at the end of eight weeks in either group, and there were no significant differences between the two groups.

### 3.3. Biochemical Characteristics of the Participants at Baseline and after 8 Weeks

Compared with those at baseline, fasting insulin, HOMA-IR, total cholesterol, LDL-C, and HDL-C were significantly lower, and FGI and QUICKI were significantly greater at 8 weeks in the vitamin D3- and probiotic-enriched OJ groups (Table 4, Figure 2E,F). No significant differences in triglyceride, CRP, IL-6, or urea levels were detected between the 8-week and baseline groups (*p* for all >0.05, Table 4). Compared with baseline, fasting glucose was significantly lower and serum vitamin D was significantly greater at 8 weeks in the control group (Table 4). A significant interaction between intervention and time was found only for the changes in total cholesterol and LDL-C in favor of the intervention group (*p* < 0.1 for all, Table 4, Figure 2E,F). These interactions did not materially change after further adjustment for BMI, medication use, fat mass, HOMA-IR, IPAQ, QUICKY, and IL-6 levels (*p* > 0.1 for all). Vitamin D levels increased significantly in the control group at 8 weeks compared to baseline, but there were no differences between the groups after adjustments for age and sex (Table 4). Serum vitamin D levels were negatively correlated with vitamin D status (rho = −0.796, *p* < 0.001) and BMI (rho = −0.364, *p* = 0.009).

### 3.4. Peripheral and Central BP at Baseline and at the End of the 8-Week Study Period

Peripheral SBP was significantly reduced at the end of eight weeks in both groups (Table 5). Peripheral DBP was significantly reduced at the end of 8 weeks only in the vitamin D3- and probiotic-enriched OJ groups, and PWV was reduced only in the control group (Table 5). Central systolic BP, central diastolic BP, and the % augmentation index did not change significantly at the end of 8 weeks in either group (Table 5). No significant interaction effect between intervention and time was found for the changes in any vascular marker between the two groups (Table 5).

## 4. Discussion

This 8-week randomized, placebo-controlled trial investigated the short-term effects of vitamin D3 and probiotics [*Lacticaseibacillus casei* Shirota and *Lactobacillus rhamnosus GG*] encapsulated in a WPI:GA matrix]-enriched OJ on cardiometabolic health. The intervention led to significant improvements in energy and macronutrient intake, body weight, insulin sensitivity, and blood lipid profiles. These findings resonate with earlier studies demonstrating the benefits of vitamin D3 and probiotics on metabolic health [1,2,3,4,5,6]. The observed synergistic effect highlights the potential of combining these bioactive compounds in a dietary matrix.

In this study, we noted a notable decrease in both the body weight and fat mass among participants who consumed OJ supplemented with vitamin D3 and probiotics despite adherence to a Westernized diet. This observation is noteworthy because it suggests the potential of fortified OJ to influence energy and macronutrient intake, thereby contributing to weight management. Our findings corroborate the hypothesis that a specific combination of vitamin D3 and probiotic strains, specifically *Lacticaseibacillus casei* Shirota and *Lacticaseibacillus rhamnosus* GG, can mitigate obesity-related factors, which is especially significant given that the participants maintained their usual dietary habits and that physical activity did not increase.

Consistent with earlier acute and short-term, randomized controlled trials, our results suggest that co-supplementation with vitamin D3 and probiotics can effectively modulate hunger and appetite, particularly in overweight individuals [9,10]. This effect might be attributed to the role of short-chain fatty acids (SCFAs) in activating G-protein-coupled receptors, which are known to enhance the secretion of anorectic hormones such as glucagon-like peptide-1 (GLP-1) and peptide YY, thus contributing to appetite suppression [29,30]. The divergent findings in the literature concerning the influence of probiotics on adiponectin and leptin levels [31,32,33] highlight the complexity of their effects on metabolic regulation.

The observed reduction in body weight in the intervention group in our study was notably associated with improved insulin metrics, including decreased fasting insulin and insulin resistance, as well as increased insulin sensitivity, when OJ enriched with vitamin D3 and probiotics was consumed. These improvements resonate with other research findings and point to the potential metabolic benefits of such fortified dietary options. Our study partially corroborates findings from research on women with gestational diabetes, where co-supplementation with vitamin D and probiotics led to significant reductions in fasting glucose and serum insulin levels, thereby enhancing insulin sensitivity [34]. Similar outcomes were observed in patients with type 2 diabetes, in which a 12-week regimen yielded substantial metabolic improvements [35]. A recent umbrella of 37 interventional meta-analyses reported that vitamin D supplementation significantly decreased fasting blood glucose, HbA1c, insulin, and HOMA, mainly in individuals with type 2 diabetes and women with gestational diabetes, as well as resulted in vitamin D dosages lower than or equal to 4000 IU [1]. The antidiabetic effects of probiotics include reducing pro-inflammatory cytokines via the NF-κB pathway, reducing intestinal permeability, and lowering oxidative stress [30]. However, our results only partially correspond with broader literature reviews and meta-analyses [33,36,37,38,39,40,41,42,43], thereby suggesting that the benefits of such supplementation may vary based on individual characteristics, including the baseline vitamin D status. It should also be noted that the results from two meta-analyses of prospective cohorts and RCTs reported no association between the consumption of fruit juices and type 2 diabetes prevalence, and there were no significant effects on glycemic control and blood glucose metabolism [44,45]. 

Our study also revealed reductions in peripheral blood pressure without corresponding changes in central blood pressure or arterial stiffness, as measured by PWC. The lack of change in central hemodynamics could be attributed to the intervention’s short duration [46]. Nonetheless, the reduction in peripheral blood pressure was consistent with previous findings on the cardiovascular benefits of fruit juice consumption [47,48], particularly in lowering systolic BP [49,50,51]. Our results are in partial agreement with a meta-analysis reporting that probiotic consumption significantly decreased systolic BP by −3.56 mmHg and diastolic BP by −2.38 mmHg compared to those in control groups, with a greater reduction in BP found with multiple probiotics compared to single probiotics, as well as greater reductions being found in diastolic BP in hypertensive patients [52]. An inverse relationship between vitamin D levels and PWV, independent of traditional risk factors, has been established [53]. Nutritional vitamin D supplementation can significantly improve arterial elasticity by reducing the PWV in adults with vitamin D deficiency [54]. However, 50% of our study population had normal vitamin D levels, and only very few participants were vitamin D-deficient, which may explain the absence of an effect of OJ on arterial elasticity.

In the present study, the group that consumed OJ supplemented with vitamin D3 and probiotics demonstrated a notable decrease in LDL-C levels compared to the control group, thus suggesting an increase in the lipid-lowering ability of the beverage. This observation aligns partially with others [55], showing that OJ reduces LDL-C levels in hypercholesterolemic adults without affecting HDL-C or triglycerides. Those authors also noted that OJ enhanced the ability of HDL to assimilate free cholesterol, which is key for cholesterol efflux and reverse transport, thereby suggesting a potential benefit to cardiovascular health [55]. Furthermore, our findings correlate to some extent with those of others [34], where vitamin D and probiotic co-supplementation in gestational diabetes led to a reduction in triglyceride levels and an improvement in lipid profiles and antioxidant status. Similarly, another study revealed improvements in lipid profiles following the consumption of pomegranate juice enriched with *Lactobacillus rhamnosus* GG in women with PCOS [48]. Our results also echo a meta-analysis of 30 RCTs with 1624 participants showing that probiotics can decrease total cholesterol and LDL-C levels, especially in individuals with high baseline cholesterol levels [6]. However, our study differs from others [42], which reported only minor improvements in total cholesterol and no significant LDL-C changes with prebiotic, probiotic, or symbiotic use in patients with type 2 diabetes. These mixed outcomes underscore the nuanced role of functional foods and supplements in lipid metabolism management. This variance could be due to the complex interplay of bioactive compounds with individual metabolic pathways and suggests a need for personalized approaches to nutritional interventions. Although our findings support the lipid-modulating potential of vitamin D3 and probiotics, they also highlight the variability of responses to such interventions and the importance of considering individual baseline health status.

Contrary to our expectations, vitamin D levels did not increase more in the vitamin D and probiotic-enriched OJ compared to the control OJ, which was not explained by vitamin D intake from the 3-day diaries or the sun exposure questionnaire. A variety of factors may reduce vitamin D absorption, including obesity, problems with absorption, or the ability to convert vitamin D to its active form, as well as variations in gut microbiota composition [56]. Moreover, fat enhances the absorption of vitamin D, a fat-soluble vitamin; therefore, using orange juice, which is fat-free, as a vehicle for vitamin D enrichment might not be optimal. Enriching the OJ with fat in the encapsulation process could potentially improve vitamin D absorption. Our findings prompt a re-evaluation of the effectiveness with which vitamin D is absorbed from fortified OJs that are commercially available. 

Although this study offers promising insights into the role of functional foods in cardiometabolic health, its limitations include the short intervention duration. Moreover, despite the randomized control methodology used in our study, differences between the groups were observed for some metabolic markers. Importantly, the interaction between intervention and time did not change after further adjustment for the HOMA-IR and QUICKI scores. Future research should aim to replicate these findings in larger, more diverse populations, and it should assess the long-term sustainability of the observed effects. The individual variability in response to the intervention also warrants further investigation, considering the influence of factors such as the gut microbiota composition. Finally, residual confounding, despite the use of multivariable linear mixed models and additional sensitivity analyses, cannot be excluded. Finally, the lack of individual vitamin D3 or probiotic groups in the study design prevents drawing conclusions about their synergistic effects, and it remains uncertain whether co-supplementation is superior to individual therapies. 

## 5. Conclusions

In conclusion, the results of this study underscore the potential of OJ enriched with vitamin D3 and probiotics (*Lacticaseibacillus casei* Shirota and *Lacticaseibacillus rhamnosus* GG) as a functional food that positively impacts metabolic health markers in adults at high cardiometabolic risk. These results encourage further exploration of the integration of functional foods into personalized nutrition strategies to mitigate metabolic dysfunction.

## Figures and Tables

**Figure 1 nutrients-16-01331-f001:**
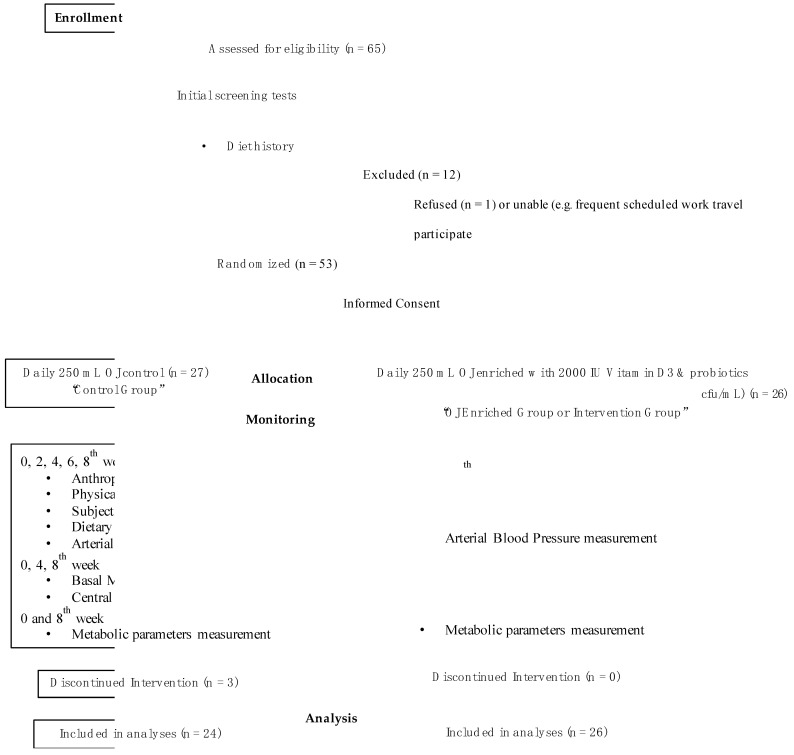
Trial design diagram showing participant flow. Abbreviations—OJ: orange juice (Aspis SA Hellenic Juice Industry, Greece). Metabolic parameters: blood glucose, insulin, blood lipids (total cholesterol, LDL-C, HDL-C, triglycerides), *C*-reactive protein, interleukin-6, urea, complete blood count, and serum vitamin D.

**Figure 2 nutrients-16-01331-f002:**
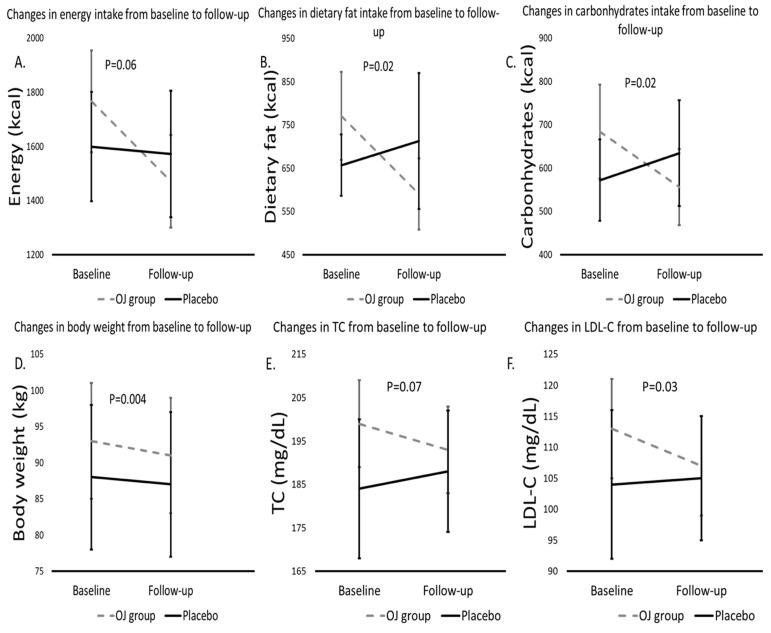
Changes in the energy (**A**) and macronutrient intake (fat: (**B**), carbohydrates: (**C**)), anthropometric indices (**D**), and biochemical characteristics (TC: (**E**), LDL-C: (**F**)) between baseline and follow-up in patients in the vitamin D3 and probiotic-enriched orange juice (OJ) group vs. those in the placebo group. The *p*-values were derived through comparisons between the enriched OJ and control OJ, adjusted for age and sex, using mixed linear models. Abbreviations: TC: total cholesterol; LDL-C: low density lipoprotein cholesterol.

**Table 1 nutrients-16-01331-t001:** The participants’ baseline characteristics (n = 50).

	Enriched OJ with Vitamin D3 and Probiotics Group (n = 26)	Conventional OJ (Control) Group (n = 24)	*p*-Value
Gender			
Male, n (%)	10 (39%)	7 (29%)	0.35
Age, years (SD)	49 ± 2	48 ± 2	0.71
Height, cm (SD)	169 ± 2	166 ± 2	0.43
Drug treatment			
No drug treatment	17 (65%)	13 (54%)	0.65
Antihypertensives, n (%)	2 (8%)	1 (4%)	
Antihyperlipidemic, n (%)	3 (12%)	3 (13%)	
Antihypertensive and antihyperlipidemic, n (%)	0 (0%)	1 (4%)	
Alcohol consumption, n (%)	17 (65%)	14 (58%)	0.32
Smoking, n (%)	17 (65%)	14 (58%)	0.84
Physical activity level, n (%)			
Low	18 (69%)	8 (33%)	0.02
Moderate	6 (23%)	8 (33%)	
High	2 (8%)	8 (33%)	

Other medical treatment in the enriched OJ vs. Control OJ groups, n (%): Hyperlipidemia: 15 (58%) vs. 12 (50%). Prediabetes: 10 (39%) vs. 10 (42%). Hypertension: 10 (39%) vs. 7 (29%).

**Table 2 nutrients-16-01331-t002:** Energy, macronutrient, and selected micronutrient intakes at baseline and at the end of 8 weeks after the consumption of vitamin D3 and probiotic-enriched OJ or control OJ (n = 50).

	Enriched with Vitamin D3 and Probiotics OJ Group (n = 26)			Control OJ Group (n = 24)				Interaction (Intervention × Time)		
	Baseline	8 Weeks	*p* ^a^	Baseline	8 Weeks	*p* ^a^	*p* ^b^	Coef.	95% CI	*p* ^0–^ ^8^
Energy (kcal)	1767 ± 94	1471 ± 86	0.002	1599 ± 101	1572 ± 117	0.83	0.23	−269.8	(−553.2, 13.7)	0.06
Protein (g)	75 ± 4	65 ± 5	0.02	73 ± 6	65 ± 5	0.18	0.82	−1.4	(−15.9, 13.1)	0.85
Protein (kcal)	300 ± 17	259 ± 19	0.02	294 ± 23	258 ± 21	0.18	0.82	−5.7	(−63.9, 52.5)	0.85
Dietary fat (g)	86 ± 6	66 ± 5	0.01	73 ± 4	79 ± 9	0.53	0.08	−26.4	(−48.9, −3.9)	0.02
Dietary fat (kcal)	771 ± 52	590 ± 42	0.01	657 ± 36	713 ± 80	0.53	0.08	−237.6	(−440.3, −34.9)	0.02
Carbohydrates (g)	162 ± 13	139 ± 11	0.04	143 ± 12	159 ± 15	0.25	0.28	−38.7	(−70.6, −6.8)	0.02
Carbohydrates (kcal)	684 ± 54	556 ± 44	0.04	572 ± 47	634 ± 61	0.25	0.28	−154.8	(−282.4, −27.3)	0.02
Dietary fiber (g)	14 ± 1	15 ± 3	0.84	13 ± 1	12 ± 1	0.53	0.45	1.7	(−5.3, 8.8)	0.63
Sugars (g)	56 ± 5	56 ± 5	0.91	51 ± 6	67 ± 6	0.02	0.53	−16.1	(−31.1, −1.0)	0.04
Saturated fat (g)	23 ± 2	20 ± 2	0.16	21 ± 2	24 ± 3	0.39	0.52	−5.4	(−11.9, 1.1)	0.11
	**Enriched with vitamin D3 and probiotics OJ group (n = 26)**			**Control OJ group (n = 24)**				**Interaction (intervention × time)**		
	**Baseline**	**8 weeks**	** *p* ^a^ **	**Baseline**	**8 weeks**	** *p* ^a^ **	** *p* ^b^ **	**Coef.**	**95% CI**	** *p* ^0–8^ **
Trans fat (g)	0 (0, 2)	0 (0, 0)	0.05	0 (0, 0)	0 (0, 0)	0.65	0.23	0.2	(−0.1, 0.3)	0.16
Monounsaturated fat (g)	32 ± 3	25 ± 2	0.02	24 ± 2	24 ± 3	0.90	0.04	−7.2	(−15.3, 0.8)	0.08
Polyunsaturated fat (g)	12 (7, 19)	8 (5, 10)	0.002	9.7 (7, 15)	7 (5, 11)	0.08	0.53	39.3	(−44.2, 122.9)	0.36
Dietary cholesterol (g)	125 ± 16	137 ± 18	0.54	153 ± 25	135 ± 29	0.57	0.35	30.5	(−39.9, 101.0)	0.40
Vitamin C (mg)	88 ± 14	104 ± 8	0.25	51 ± 7	88 ± 9	0.002	0.02	−21.6	(−55.2, 11.99)	0.21
Salt (g)	4.9 ± 0.5	3.8 ± 0.3	0.02	4.0 ± 0.3	3.9 ± 0.4	0.71	0.15	−0.9	(−2.1, 0.3)	0.15
Sodium (mg)	1839 ± 185	1436 ± 125	0.03	1544 ± 121	1450 ± 153	0.59	0.19	−308.3	(−780.2, 163.6)	0.20
Potassium (mg)	1917 ± 98	1762 ± 91	0.15	1591 ± 136	1778 ± 89	0.19	0.06	−342.8	(−672.7, 12.8)	0.04

The normally distributed variables are the means ± SEMs, and the skewed variables are the median (1st and 3rd quartiles). The *p*^a^-values describe the significant difference in each group compared to baseline, and they were obtained with a dependent paired sample *t*-test for the normally distributed variables and a Wilcoxon test for the nonparametric data. The *p*^b^-values were derived from comparisons at baseline between the vitamin D3 and probiotic-enriched orange juice and control orange juice by using an independent sample *t*-test for the normally distributed variables and the Mann–Whitney U test for the nonparametric variables. The *p*^0–8^-values were derived through comparisons between the vitamin D3 and probiotic-enriched orange juice and control orange juice adjusted for age and gender by using mixed linear models.

**Table 3 nutrients-16-01331-t003:** The anthropometric characteristics, body composition analysis, and basal metabolic rate (BMR) at baseline and 8 weeks after the consumption of vitamin D3 and probiotic-enriched OJ or control OJ (n = 50).

	Enriched with Vitamin D3 and Probiotics OJ Group (n = 26)			Control OJ Group (n = 24)				Interaction Intervention × Time (0–8 weeks)		
	Baseline	8 Weeks	*p* ^a^	Baseline	8 Weeks	*p* ^a^	*p* ^b^	Coef.	95% CI	*p* ^0–8^
Body weight (kg)	93 ± 4	91 ± 4	<0.001	88 ± 5	87 ± 5	0.22	0.38	−1.4	(−2.4, −0.4)	0.004
BMI (kg/m^2^)	33 ± 1	32 ± 1	0.005	32 ± 1	31 ± 1	0.14	0.51	−0.06	(−1.1, −0.1)	0.01
Body fat mass (kg)	38 ± 2	36 ± 2	0.003	36 ± 2	35 ± 3	0.01	0.54	−0.5	(−1.6, 0.6)	0.38
Body water (kg)	41 ± 2	39 ± 2	0.28	38 ± 2	38 ± 2	0.84	0.46	−1.9	(−5.0, −1.2)	0.23
Muscle mass (kg)	28 (25)	28 (24)	0.45	26 (23)	26 (24)	0.32	0.43	−0.3	(−0.7, 0.2)	0.27
Lean body mass (kg)	55 ± 3	55 ± 3	0.26	52 ± 3	53 ± 3	0.27	0.46	−0.5	(−1.3, 0.3)	0.18
Waist circumference (cm)	108 ± 4	103 ± 3	0.10	101 ± 4	101 ± 4	0.95	0.18	−4.7	(−10.4, 1.1)	0.11
Hip circumference (cm)	116 ± 2	113 ± 2	<0.001	112 ± 2	112 ± 2	0.76	0.13	−4.2	(−9.1, 0.6)	0.08
Waist-to-hip circumference ratio (cm)	0.9 ± 0.03	0.9 ± 0.02	0.67	0.9 ± 0.03	0.89 ± 0.02	0.64	0.56	−0.005	(−0.07, 0.07)	0.91
BMR (kcal/day)	1489 ± 109	1504 ± 95	0.85	1334 ± 87	1501 ± 92	0.04	0.28	−156.6	(−368.9, 55.7)	0.15

The normally distributed variables are the means ± SEMs, and the skewed variables are the median (1st and 3rd quartiles). The *p*^a^-values describe the significant difference in each group compared to baseline and were obtained with a dependent sample *t*-test for the normally distributed variables and a Wilcoxon test for the nonparametric data. The *p*^b^-values were derived from comparisons at baseline between the vitamin D3 and probiotic-enriched orange juice and control orange juice by using an independent sample *t*-test for the normally distributed variables and the Mann–Whitney U test for the nonparametric variables. The *p*^0–8^-values were derived through comparisons between the vitamin D3 and probiotic-enriched orange juice and control orange juice adjusted for age and gender by using mixed linear models.

**Table 4 nutrients-16-01331-t004:** The biochemical characteristics of the participants at baseline and at the end of 8 weeks after the consumption of vitamin D3 and probiotic-enriched OJ or control OJ (n = 50).

	Enriched with Vitamin D3 and Probiotics OJ Group (n = 26)			Control OJ Group (n = 24)				Interaction 0–8 Weeks		
	Baseline	8 Weeks	*p* ^a^	Baseline	8 Weeks	*p* ^a^	*p* ^b^	Coef.	95% CI	*p* ^0–8^
Fasting glucose (mg/dL)	94 ± 2	91 ± 2	0.12	90 ± 2	87 ± 2	0.02	0.18	−0.6	(−5.1, 4.0)	0.80
Fasting insulin (μU/mL)	9 (7, 12)	8 (6, 11)	0.04	7 (4, 9)	6 (4, 8)	0.45	0.01	−1.5	(−3.9, 0.8)	0.21
HOMA-IR	2.2 (1.4, 2.9)	2 (1.2, 2.5)	0.02	1.4 (1.0, 2.2)	1.2 (0.8, 1.7)	0.28	0.02	−0.4	(−0.9, 0.2)	0.23
FGI	11 ± 0.9	13 ± 2	0.03	15 ± 1.5	16 ± 1.5	0.66	0.02	−2.1	(−1.0, 5.1)	0.19
QUICKI	0.3 ± 0.01	0.4 ± 0.01	0.006	0.4 ± 0.01	0.4 ± 0.01	0.23	0.01	0.007	(−0.01, 0.02)	0.27
Total cholesterol (mg/dL)	199 ± 5	193 ± 5	0.05	184 ± 8	188 ± 7	0.45	0.13	−10.3	(−21.4, 0.9)	0.07
LDL-C (mg/dL)	113 ± 4	107 ± 4	0.004	104 ± 6	105 ± 5	0.77	0.22	−6.9	(−13.5, −0.5)	0.03
HDL-C (mg/dL)	50 ± 2	47 ± 2	0.01	51 ± 2	50 ± 2	0.85	0.84	−2.5	(−6.4, 1.3)	0.20
Triglycerides (mg/dL)	107 ± 10	105 ± 10	0.65	89 ± 11	90 ± 9	0.83	0.22	−4.1	(−21.2, 13.0)	0.64
CRP (mg/dL)	2 (1, 4)	2 (1, 6)	0.87	2 (1, 3)	2 (1, 3)	0.94	0.61	−0.03	(−0.8, 0.8)	0.95
IL-6 (pg/mL)	2 (2, 3)	2 (2, 3)	0.55	2 (2, 2)	2 (2, 3)	1.00	0.14	0.05	(−0.5, 0.6)	0.87
Vitamin D (ng/mL)	20 ± 2	21 ± 1	0.29	22 ± 2	25 ± 2	0.007	0.45	−1.9	(−4.8, 1.0)	0.19
Urea (mg/dL)	28 ± 1	28 ± 2	0.79	31 ± 3	32 ± 2	0.52	0.23	−0.7	(−4.5, 3.2)	0.74

The normally distributed variables are the means ± SEMs, and the skewed variables are the median (1st and 3rd quartiles). Abbreviations: HOMA-IR = homeostasis model assessment for insulin resistance, FGI = fasting-glucose-to-insulin ratio, QUICKI = quantitative insulin sensitivity check index, LDL = low-density lipoprotein, HDL = high-density lipoprotein, CRP = C reactive protein, IL-6 = Interleukin-6. The *p*^a^-values describe the significant difference in each group compared to baseline and were obtained with a dependent sample *t*-test for the normally distributed variables and a Wilcoxon test for the nonparametric data. The *p*^b^-values were derived from comparisons at baseline between the vitamin D3 and probiotic-enriched orange juice and control orange juice by using an independent sample *t*-test for the normally distributed variables and a Mann–Whitney U test for the nonparametric variables. The *p*^0–8^-values were derived through comparisons between the vitamin D3 and probiotic-enriched orange juice and control orange juice adjusted for age and gender by using mixed linear models.

**Table 5 nutrients-16-01331-t005:** Peripheral and central blood pressure, augmentation index, and pulse wave velocity of participants at baseline and at the end of 8 weeks after the consumption of vitamin D3 and probiotic-enriched OJ or control OJ (n = 50).

	Enriched with Vitamin D3 and Probiotics OJ Group (n = 26)			Control OJ Group (n = 24)				Interaction 0–8 Weeks		
	Baseline	8 Weeks	*p* ^a^	Baseline	8 Weeks	*p* ^a^	*p* ^b^	Coef.	95% CI	*p* ^0–8^
SBP (mmHg)	127 ± 3	120 ± 2	0.002	124 ± 2	118 ± 2	**0.01**	0.48	−0.0	(−6.8, 5.1)	0.77
DBP (mmHg)	82 ± 3	77 ± 2	0.02	79 ± 2	77 ± 2	0.43	0.27	−4.1	(−9.5, 1.3)	0.14
Central SBP (mmHg)	117 ± 3	116 ± 4	0.74	120 ± 5	113 ± 3	0.25	0.57	5.5	(−7.1, 18.1)	0.39
Central DBP (mmHg)	78 ± 2	77 ± 2	0.60	78 ± 2	76 ± 2	0.57	0.93	0.5	(−4.3, 5.3)	0.84
Augmentation index, Alx (%)	4 (−19, 21)	6 (−8.8, 12.3)	0.75	12 (−35, 30)	7 (−36, 13.5)	0.22	0.20	16.2	(−8.8, 41.2)	0.21
PWV (m/s)	8 (7, 9)	8 (7, 9)	0.82	8 (8, 10)	8 (7.3, 8)	0.03	0.12	16.2	(−8.9, 41.2)	0.26

The normally distributed variables are the means ± SEMs, and the skewed variables are the median (1st and 3rd quartiles). Abbreviations: SBP = systolic blood pressure, DBP = diastolic blood pressure, and PWV = pulse wave velocity. The *p*^a^-values describe the significant difference in each group compared to baseline and were obtained with a dependent sample *t*-test for the normally distributed variables and a Wilcoxon test for the nonparametric data. The *p*^b^-values were derived from comparisons at baseline between the vitamin D3 and probiotic-enriched orange juice and control orange juice by using an independent sample *t*-test for the normally distributed variables and a Mann–Whitney U test for the nonparametric variables. The *p*^0−8^-values were derived through comparisons between the vitamin D3 and probiotic-enriched orange juice and control orange juice adjusted for age and gender by using mixed linear models.

## Data Availability

The original contributions presented in the study are included in the article, further inquiries can be directed to the corresponding author.

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
