# Peer review of "Two-Month Consumption of Orange Juice Enriched with Vitamin D3 and Probiotics Decreases Body Weight, Insulin Resistance, Blood Lipids, and Arterial Blood Pressure in High-Cardiometabolic-Risk Patients on a Westernized Type Diet: Results from a Randomized Clinical Trial"

_nutrients, 2024, doi:10.3390/nu16091331_

Round 1
Reviewer 1 Report
Comments and Suggestions for Authors
The study conducted by Papakonstantinou et al., a randomized control trial, aimed to assess the effects of enriched orange juice with vitamin D3 and probiotics on various health parameters in high cardiometabolic risk patients following a westernized type diet. While the study has its merits, there are several aspects that require clarification by the authors:
1- Detailed information about the diets of the two groups and their adherence is crucial for understanding the potential impact on study outcomes.
2- The preparation of orange juices by the researchers raises questions regarding quality control aspects. Were measures taken to ensure the efficiency of the process? For instance, was the bioavailability of probiotics tested post-preparation?
3- How was the dosage of the probiotics determined? A discussion on the rationale behind the chosen dosage would enhance the understanding of the intervention.
4- Did the addition of probiotics affect the palatability and organoleptic characteristics of the orange juice? Considering the various ingredients added during preparation, it's essential to address any potential impact on taste and consistency. Moreover, the control orange juice should have been formulated similarly, without adding vitamin D and probiotics, to ensure proper blinding of participants.
5- The study design's lack of individual vitamin D or probiotics groups prevents drawing conclusions about their synergistic effects. Therefore, the conclusions should be limited to the combination therapy, as it remains uncertain whether this approach is superior to individual therapies.
6- Authors should discuss the potential benefits of non-enriched orange juice, particularly considering its significant effects on various outcomes, including systolic blood pressure.
7- How the authors explain the increase in vitamin D levels in the control group, compared to the enriched group?
8- Despite randomization, baseline differences between the treatment groups, as indicated in Table 4, suggest potential biases that may impact the final conclusions. Authors should explicitly acknowledge and address this limitation in the paper.
Author Response
Title: Two-month consumption of orange juice enriched with vitamin D3 and probiotics decreases body weight, insulin resistance, blood lipids and arterial blood pressure in high cardiometabolic risk patients on a westernized type diet. Results from a randomized clinical trial.
Detailed Responses to Reviewers’ Comments
We express our gratitude to the Editor and Reviewers for their insightful comments, which have been instrumental in the refinement of our manuscript. The manuscript has undergone thorough revisions to fully incorporate the guidance and recommendations of the Reviewers. A detailed, point-by-point response to each comment is provided, as follows:
Reviewer 1:
Comment 1: Detailed information about the diets of the two groups and their adherence is crucial for understanding the potential impact on study outcomes.
Authors’ Response: We thank the reviewer for the helpful and insightful comments. Participants were asked to maintain their usual dietary intake and habits and no diet was given to them. We have added two paragraphs in the revised manuscript to address your concerns. “Participants were asked to record the juice serving and foods and beverages they consumed, including the time and location of consumption and any additional notes that might be relevant, such as feelings of satiety or reasons for missing a serving or meal”, please see Lines 161-164, and “Compliance with daily drinking of the allocated juice and maintaining usual dietary and exercise habits was assessed using weekly food records and a physical activity questionnaire, as well as by scheduled telephone contact of dietitians with the participants in the middle of the week. During these check-ins, food diaries were reviewed, and any discrepancies or omissions were discussed. Participants were given seven portions of OJ weekly and asked to return any unopened portions at the end of each week. No unopened OJ was returned at any time point”, please see Lines 172-179.
Comment 2: The preparation of orange juices by the researchers raises questions regarding quality control aspects. Were measures taken to ensure the efficiency of the process? For instance, was the bioavailability of probiotics tested post-preparation?
Authors’ Response: We are grateful to the reviewer for the helpful and insightful comments. Orange juice samples were prepared under aseptic conditions (inside a laminar flow cabinet) using sterile bags (sterile bags for breast milk – 250mL) as juice containers. All reagents used were sterilized before use and the personnel wore the appropriate protective equipment to ensure the aseptic process. To measure the bioavailability of the probiotics in the finished products, first the encapsulated cells were released from the microcapsules by adding the appropriate amount of 5 N NaOH to increase the pH value to 7.4; at this pH, the biopolymers in the coacervate structures are no longer interacting electrostatically since they both carry negatively charged groups. Then, the probiotic population was determined by counting viable cells present in the juice samples after sampling (10-fold dilutions in ¼ strength Ringer’s solution) the juices and pour plating in MRS agar (incubation at 37°C for 48-72h) (detail information can be found at Bosnea et al., 2017, https://doi.org/10.1016/j.lwt.2016.11.056). This procedure was done immediately after juice preparation and during the 7-day cold storage. This information was also added in the revised manuscript. Please see Lines 227-239.
Comment 3: How was the dosage of the probiotics determined? A discussion on the rationale behind the chosen dosage would enhance the understanding of the intervention.
Authors’ Response: We thank the reviewer for the helpful and insightful comments. The health effects of probiotics are reliant on dose and the minimum recommended amount to be consumed is often defined as 109 CFU per day, labeled as CFU/ml or CFU/gram (Minelli and Benini, 2009). In that sense, the dosage of the 2 encapsulated probiotic strains (108 CFU/mL of each strain) was selected to achieve this amount (109 CFU per day) and the fact that a large enough number of encapsulated microorganisms also remain viable and survive the transit through the gastrointestinal tract (GIT), while facing stressors such as bile and gastric acid (the numbers of viable microorganisms required to obtain a clinical effect is generally considered to be 106 CFU/ml in the small bowel and 108 CFU/g in the colon, as reported by Minelli and Benini, 2009 and FAO/WHO, 2006). In vitro experiments simulating the GIT were undertaken in our laboratory, and it was shown that the selected encapsulated probiotic strains can survive and maintain their population at high levels (108 CFU/mL). Preliminary results can be found on EFFoST 2022 International Conference Title: Microencapsulation of probiotic cells enhances their survival under conditions simulating the human gastrointestinal system, by Stamatia Vitsou-Anastasiou, Olga S. Papadopoulou, Apostolos Karkos, Anthoula A. Argyri, Agapi Doulgeraki, George-John Nychas and Chrysoula Tassou, added in the acknowledgment section, please see lines 601-605. This information was also added in the revised manuscript, please see Lines 215-217.
Comment 4: Did the addition of probiotics affect the palatability and organoleptic characteristics of the orange juice? Considering its various ingredients added during preparation, it’s essential to address any potential impact on taste and consistency.
Authors’ Response: We are grateful to the reviewer for the helpful and insightful comments. The addition of vitamin D and encapsulated probiotics did not alter the organoleptic characteristics of the product. Vitamin D is colorless and odorless and its addition to the juice cannot be detected. In addition, several sensory assessments (by trained panel of the Institute of Technology of Agricultural Products) during cold storage of the juices (control and FJ) were undertaken prior to the clinical studies, to ensure the palatability of the product and to investigate if any impact on taste or consistency exists. Those results are not presented. This information was added in the revised manuscript, please see Lines 219-226.
Comment 5: Moreover, the control orange juice should have been formulated similarly, without adding vitamin D and probiotics, to ensure proper blinding of participants.
Authors’ Response: We thank the reviewer for the helpful and insightful comments. Orange juice (control and FJ) portions were prepared using sterilized breast milk bags of 250 mL and volunteers were not aware of the group that were participating. The encapsulated probiotics and vit D were not visible to the volunteers and no deviation in the taste or in the palatability of the juices was evident. This information was added in the revised manuscript, please see Lines 227-229.
Comment 6: The study design’s lack of individual vitamin D or probiotics groups prevents drawing conclusions about their synergistic effects. Therefore, the conclusions should be limited to the combination therapy, as it remains uncertain whether this approach is superior to individual therapies.
Authors’ Response: We are grateful to the reviewer for the helpful and insightful comments. We have added the reviewer’s comment in the limitations section of the revised manuscript, please see Lines 565-568, and we have modified the conclusions section, please see Lines 570-572.
Comment 6: Authors should discuss the potential benefits of non-enriched orange juice, particularly considering its significant effects on various outcomes, including systolic blood pressure.
Authors’ Response: We thank the reviewer for the helpful and insightful comments. We have added information regarding the potential benefits of non-enriched orange juice in the revised discussion section, please see Lines 500-503 and 509.
Comment 7: How the authors explain the increase in vitamin D levels in the control group, compared to the enriched group?
Authors’ Response: We are grateful to the reviewer for the helpful and insightful comment. There were no significant differences between groups after the required adjustments regarding serum vitamin D levels. However, the reviewer is correct that we failed to show an improvement in serum vitamin D levels in the OJ enriched group compared to the control group. We added a paragraph discussing bioavailability issues and added information and results from the vitamin D questionnaire that we used and vitamin D intake from the 3-day diaries in the revised manuscript, please see Lines 263-273 and 379-385 and 427-431 and 517-518 and 544-554.
Comment 8: Despite randomization, baseline differences between the treatment groups, as indicated in Table 4, suggest potential biases that may impact the final conclusions. Authors should explicitly acknowledge and address this limitation in the paper.
Authors’ Response: We thank the reviewer for the helpful and insightful comment. We have added the suggested comment in the revised limitations section, please Lines 556-560.

Reviewer 2 Report
Comments and Suggestions for Authors
the article is well written, I only have a few small comments:
- line 38: caloric intake? caloric expenditure?
- line 61: double dot
- line 63: What type of probiotic?
- line 67: no explanation of abbreviations in the main text
- line 80: In healthy individuals?
- line 148: repetition of the manufacturer's name
- figure 1: placing the figure and caption on the same page makes it easier to read; part of the text (the lower part of the figure) is cut away
- line 203: Were other concentrations of probiotics tested?
- line 225: Has the amount of vitamin D and/or probiotics in the finished product been confirmed/verified?
- line 255: HDL-"C" - capital letter
- line 268: looks like a different font
- table 1: description of the table should not be part of it; a smaller interline will help reduce the size of the table; "other medical treatment" - please list in a table footnote
- table 2: lack of continuity of column lines; if the table takes up several pages the repetition of headings makes it easier to read
- table 3: pa column should be wider
- line 372: unnecessary dot
Author Response
Title: Two-month consumption of orange juice enriched with vitamin D3 and probiotics decreases body weight, insulin resistance, blood lipids and arterial blood pressure in high cardiometabolic risk patients on a westernized type diet. Results from a randomized clinical trial.
Detailed Responses to Reviewers’ Comments
We express our gratitude to the Editor and Reviewers for their insightful comments, which have been instrumental in the refinement of our manuscript. The manuscript has undergone thorough revisions to fully incorporate the guidance and recommendations of the Reviewers. A detailed, point-by-point response to each comment is provided, as follows:
Reviewer #2:
The article is well written, I only have a few small comments:
Comment 1: line 38: caloric intake? caloric expenditure?
Authors’ Response: We thank the reviewer for noticing this and we apologize for the confusion we may have caused. We have replaced “caloric” with “energy” intake to help with content clarity, in the revised abstract, please see Line 39.
Comment 2: line 61: double dot.
Authors’ Response: We thank the reviewer for noticing this. We have removed the double dot in the revised introduction section, please see Line 64.
Comment 3: line 63: What type of probiotic?
Authors’ Response: We thank the reviewer for the helpful comment. We have added the type of probiotic, lactic bacteria, please see Line 66.
Comment 4: line 67: no explanation of abbreviations in the main text.
Authors’ Response: We are grateful to the reviewer for noticing this. We have explained the abbreviation in the revised text, please see Line 70.
Comment 5: line 80: in healthy individuals?
Authors’ Response: We thank the reviewer for the helpful and insightful comment. We added the “healthy individuals” in the revised introduction, please see Line 84.
Comment 6: line 148: repetition of the manufacturer’s name.
Authors’ Response: We thank the reviewer for noticing this. We have removed the repetition of the manufacturer’ name in the revised text.
Comment 7: figure 1: placing the figure and caption on the same page makes it easier to read; part of the text (the lower part of the figure) in cut away.
Authors’ Response: We are grateful to the reviewer for the helpful and insightful comment. We have made all suggested modifications, please see the revised Figure 1.
Comment 8: Were other concentrations of probiotics tested?
Authors’ Response: We thank the reviewer for the helpful and insightful comment. To select the appropriate concentration of the probiotics, several sensory assessments (by trained panel of the Institute of Technology of Agricultural products) testing different concentrations in the probiotic population were undertaken prior to the clinical studies. We concluded that the 8 log CFU/mL did not affect the palatability of the product and did not have any impact on taste or consistency. Also, since the health effects of probiotics are reliant on dose and the minimum recommended amount to be consumed is often defined as 109 CFU per day, the selected dose fulfills this criterion. Please see Lines, 215-218.
Comment 9: Has the amount of vitamin D and/or probiotics in the finished product been confirmed/verified?
Authors’ Response: We thank the reviewer for the helpful and insightful comment. Vit D3 was randomly checked using HPLC method and no deviations from the initial concentration were found during the 7-day storage. To measure the bioavailability of the probiotics in the finished products, first the encapsulated cells were released from the microcapsules by adding the appropriate amount of 5 N NaOH to increase the pH value to 7.4; at this pH, the biopolymers in the coacervates are no longer interacting electrostatically, since they both carry negatively charged groups. Then, the probiotic population was determined by counting viable cells present in the juice samples after sampling (10-fold dilutions in ¼ strength Ringer’s solution) the juices and pour plating in MRS agar (incubation at 37°C for 48-72h) (detail information can be found at Bosnea et al., 2017, https://doi.org/10.1016/j.lwt.2016.11.056). This procedure was done immediately after juice preparation and during the 7-day cold storage. Please see Lines 219-226.
Comment 10: line 255: HDL-“C” – capital letter
Authors’ Response: We thank the reviewer for noticing this. We corrected it as suggested, please see Line 280.
Comment 11: line 268: looks like a different font
Authors’ Response: We thank the reviewer for noticing this. We made the suggested modifications, please see Lines 293-299.
Comment 12: table 1: description of the table should not be part of it; a smaller interline will help reduce the size of the table; “other medical treatment” – please list in a table footnote.
Authors’ Response: We thank the Reviewer for this suggestion. We have decreased the interline in all tables and have moved the description of the table above each table. We have also listed medical treatment in a table footnote, as recommended. Please see the revised Table 1, Line 362.
Comment 13: lack of continuity of column lines; if the table takes up several pages the repetition of headings makes it easier to read.
Authors’ Response: We appreciate the reviewer’s feedback. We have added the headings Tables extending over one page. Appropriate changes in Table 2 have been made. Please see the revised Table 2.
Comment 14: table 3: pa column should be wider.
Authors’ Response: We thank the reviewer for the helpful comment. We have made appropriate changes, please see the revised Table 3.
Comment 15: line 372: unnecessary dot.
Authors’ Response: We thank the reviewer for noticing this. We have removed the dot in the revised manuscript.
